# TEXTUAL DECOMPOSITION THEN SUB-MOTION-SPACE SCATTERING FOR OPEN-VOCABULARY MOTION GENERATION

## ABSTRACT

Text-to-motion generation is a crucial task in computer vision, which generates the target 3D motion by the given text. The existing annotated datasets are limited in scale, resulting in most existing methods overfitting to the small datasets and unable to generalize to the motions of the open domain. Some methods attempt to solve the open-vocabulary motion generation problem by aligning to the CLIP space or using the Pretrain-then-Finetuning paradigm. However, the current annotated dataset's limited scale only allows them to achieve mapping from sub-text-space to sub-motion-space, instead of mapping between full-text-space and full-motion-space (full mapping), which is the key to attaining open-vocabulary motion generation. To this end, this paper proposes to leverage the atomic motion (simple body part motions over a short time period) as an intermediate representation, and leverage two orderly coupled steps, i.e., Textual Decomposition and Sub-motion-space Scattering, to address the full mapping problem. For Textual Decomposition, we design a fine-grained description conversion algorithm, and combine it with the generalization ability of a large language model to convert any given motion text into atomic texts. Sub-motion-space Scattering learns the compositional process from atomic motions to the target motions, to make the learned sub-motion-space scattered to form the full-motion-space. For a given motion of the open domain, it transforms the extrapolation into interpolation and thereby significantly improves generalization. Our network, **DSO**-Net, combines textual **d**ecomposition and sub-motion-space **s**cattering to solve the **o**pen-vocabulary motion generation. Extensive experiments demonstrate that our DSO-Net achieves significant improvements over the state-of-the-art methods on open-vocabulary motion generation.

## 1 INTRODUCTION

Text-to-motion (**T2M**) generation, aiming at generating the 3D target motion described by the given text, is an important task in computer vision and has garnered significant attention in recent research. It plays a crucial role in various applications, such as robotics, animations, and film production.

Benefiting from advancements in GPT-style (e.g., LlamaGen (Sun et al., 2024)) and diffusion-style generative paradigm in text-to-image and text-to-video domains, some studies (Zhang et al., 2023a; Jiang et al., 2023; Tevet et al., 2022b; Shafir et al., 2023) have started using these technologies to address the T2M generation task. During the training process, paired text-motion data are utilized to align the text space with the motion space. However, the **open-vocabulary text-to-motion generation** remains a challenging problem, requiring good motion generation quality for unseen open-vocaulary text at inference. Due to the limited scale of recent high-quality annotated datasets (e.g., KIT-ML (Plappert et al., 2016) and HumanML3D (Guo et al., 2022)), as illustrated in Fig. 1 top-left, the Simple Mapping paradigm only learns a mapping between a limited sub-text-space and a sub-motion-space, rather than the mapping from full-text-space to full-motion-space. Consequently, generalization to unseen open-vocabulary text is almost impossible.

To enhance the model's generalization capabilities, two main strategies have been explored. As shown in Fig. 1 top-right, the first paradigm is CLIP-based Alignment (e.g., MotionCLIP (Tevet

et al., 2022a) and OOHMG (Lin et al., 2023)). This kind of approach aims to align the motion space with both the CLIP text space (Radford et al., 2021) and the image space. The core process involves fitting the motion skeleton onto the mesh of the human body SMPL (Loper et al., 2023) model and performing multi-view rendering to obtain pose images, thereby achieving alignment between motion and image spaces. The second paradigm is Pretrain-then-Finetuning (e.g., OMG (Liang et al., 2024a)), as illustrated in Fig. 1 bottom-left. Inspired by the success of the Stable Diffusion (Rombach et al., 2022) model in the text-to-image field, this paradigm follows a pretrain-then-finetuning process, along with scaling up the model, to enable generalization to open-vocabulary text.

Although these two types of methods have achieved some progress in open-vocabulary text-to-motion generation, they suffer from inherent flaws: (1) The CLIP-based alignment paradigm aligns static poses with the image space, which results in the loss of temporal information during the learning process. Consequently, this approach generates unrealistic motion. Furthermore, this method overlooks the feature space differences between the CLIP and T2M task datasets, potentially leading to misalignment, unreliability, and inaccuracy in motion control. (2) Although the Pretrain-then-Finetuning paradigm utilizes an adequate motion prior and expands the motion space by pretraining on large-scale motion data, the annotated paired data in the finetuning stage is severely limited. The significant imbalance between labeled and unlabeled data results in the fine-tuning stage only learning the mapping from text to a **condensed subspace** of the full motion-space. Consequently, the model has to perform extrapolation and has difficulties in generating motions that are outside the subspace distribution, as shown in Fig. 1.

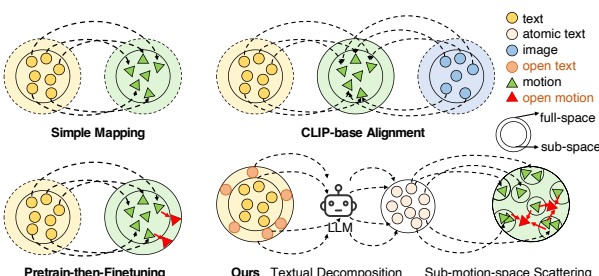

Figure 1: Compared with current text-to-motion paradigms (Simple mapping, CLIP-based alignment, and Pretrain-then-Finetuning), our method proposes the textual decomposition to decompose the raw motion text into atomic texts and sub-motion-space scattering to learn the composition process from atomic motions to target motions, which significantly improves the ability of open-vocabulary motion generation.

Consequently, we conclude that the problem of insufficient generalization ability in current methods arises from the limited amount of high-quality labeled data and the inadequate utilization of large-scale unsupervised motion data for pretraining. Existing methods can only establish overfitted mappings within a limited subspace. To achieve open-vocabulary motion generation, it is essential to establish a mapping from the full-text-space to the full-motion-space.

We observe that when understanding a motion, human beings tend to partition it into the combination of several simple body part motions (such as "spine bend forward", "left hand up") over a short time period, which we define as **atomic motions**. All these atomic motions are combined spatially and temporally to form the raw motion. This observation motivates us to **decompose a raw motion into atomic motions** and leverage atomic motions as an intermediate representation. Since raw motion texts often contain abstract and high-level semantic meanings, directly using raw texts to guide motion generation can hinder the model's ability to understand open-vocabulary motion texts. In contrast, atomic motion texts provide a low-level and concrete description of different limb movements, which are shared across different domains.

Leveraging the atomic motions as an intermediate representation, we propose to address the full-mapping problem through two orderly coupled steps: **(1) Textual Decomposition.** To enhance generalization ability, we first design a textual decomposition process that converts a raw motion text into several atomic motion texts, subsequently generating motions from these atomic texts. To prepare training data, we develop a **fine-grained description conversion** algorithm to establish atomic texts and motion pairs. Specifically, we partition the input motion into several time periods, and describe the movements of each joint and spatial relationships from the aspects of velocity (e.g., fast), magnitude (e.g., significant), and low-level behaviors (e.g., bending) for each period. The fine-grained descriptions and the raw text are then input into a large language model (LLM) to summarize the atomic motion texts. Each raw motion text is decomposed into atomic motion texts of

six body parts: the spine, left/right-upper/lower limbs, and trajectory. By this process, we guarantee the converted atomic motion texts are consistent with the actual motion behavior. During inference, for any given text, we employ the LLM to split the whole motion into several periods and describe each period using atomic motion texts. In this way, we establish a mapping from the full-text-space to the full-**atomic-text-space.(2) Sub-motion-space Scattering.** After obtaining the full-text-space to the full-**atomic-text-space** mapping in the first step, we aim to further achieve alignment from the full-atomic-text-space to the full-motion-space, through the Sub-motion-space Scattering step, thereby establishing the mapping from the full-text-space to the full-motion-space. Given the limited labeled data, it is difficult for the Pretrain-then-Finetuning paradigm to learn the full-motion-space, because its trivial alignment process only learns a mapping from text to a condensed subspace (which we refer to as sub-motion-space), requiring extrapolation for out-of-domain motions. In contrast, our approach scatters the sub-motion-space to form the full-motion-space, as shown in the bottom right of Fig. 1, transforming extrapolation into interpolation and significantly improving generalization. The sub-motion-space scattering is achieved by learning the combinational process of atomic motions to generate target motions, with a text-motion alignment (TMA) module to extract features for atomic motion texts, and a compositional feature fusion (CFF) module to fuse atomic text features into motion features and learn the the combinational process from atomic motions to target motions. As shown in Fig. 1, Interpolating an out-domain motion is essentially a combination of several nearest clusters of scattered sub-motion-space, which is highly consistent with the process of CFF we design. Therefore, the CFF ensures for scattering sub-motion-space we learned.

Overall, we adopt the discrete generative mask modeling and follow the pretrain-then-finetuning pipeline for open-vocabulary motion generation. First, we pretrain a residual VQ-VAE (Martinez et al., 2014) network using a pre-processed large-scale unlabeled motion dataset, to enable the network to have prior knowledge of large-scale motion. For the fine-tuning stages, we first leverage our textual decomposition module to convert the raw motion text into atomic texts. Then, we utilize both raw text and the atomic texts with our proposed TMA and CFF modules to train a text-to-model generative model. Our network, abbreviated as **DSO**-Net, combines textual **d**ecomposition and sub-motion-space **s**cattering to solve the **o**pen-vocabulary motion generation. We conduct extensive experiments comparing our approach with previous state-of-the-art approaches on various open-vocabulary datasets and achieve a significant improvement quantitatively and qualitatively.

In summary, our main contributions include: (1) we propose to leverage atomic motions as an intermediate representation, and design textual decomposition and sub-motion-space scattering framework to solve open-vocabulary motion generation. (2) For textual decomposition, we design a rule-based fine-grained description conversion algorithm and combine it with the large language model to obtain the atomic motion texts for a given motion. (3) For sub-motion-space scattering, we propose to leverage a text-motion alignment (TMA) module and a compositional feature fusion (CFF) module to learn the generative combination of atomic motions, thereby significantly improving the model's generalization ability.

## 2 RELATED WORKS

### 2.1 TEXT-TO-MOTION GENERATION

Text-to-Motion has been a long-standing concern. previous works (Guo et al., 2020; Petrovich et al., 2021) usually generate a motion based on the given action categories. Action2Motion (Guo et al., 2020) uses a recurrent conditional variational autoencoder (VAE) for motion generation. It uses historical data to predict subsequent postures and follows the constraints of action categories. Subsequently, ACTOR (Petrovich et al., 2021) encodes the entire motion sequence into the latent space, which significantly reduces the accumulated error. Using only action labels is not flexible enough. Therefore, some works began to explore generating motion under text (i.e., natural language). TEMOS (Petrovich et al., 2022) uses a variational autoencoder (VAE) (Kingma et al., 2019) architecture to establish a shared latent space for motion and text. This model aligns the two distributions by minimizing the Kullback-Leibler (KL) divergence between the motion distribution and the text distribution. Therefore, in the inference stage, only text input is needed as a condition to generate the corresponding motion. T2M (Guo et al., 2022) further learns a text-to-length estimator, enabling the network to give the generated motion length automatically. T2M-GPT (Zhang et al., 2023a) first introduces the VQ-VAE technique into text-to-motion tasks and leverages the autore-

gressive paradigm to generate motions. MotionGPT (Jiang et al., 2023; Zhang et al., 2024b) further improves the motion quality under the autoregressive paradigm from the aspect of text encoder. MDM (Tevet et al., 2022b) and MotionDiffuse (Zhang et al., 2022) are the first works to solve the motion synthesis task by using diffusion models. Subsequent works (Chen et al., 2023; Zou et al., 2024; Zhang et al., 2023b; Dai et al., 2024; Zhang et al., 2024a; Karunratanakul et al., 2023; Xie et al., 2023; Fan et al., 2024) further improve the controllability and quality of the generation results through some techniques such as database retrieval, spatial control, and fine-grained description. However, all these methods essentially overfit the limited training data, thereby can not achieve open-vocabulary motion generations.

## 2.2 OPEN-VOCABULARY GENERATION

Compared to the previous method of training and testing on the same dataset, the open-vocabulary generation task expects to train on one dataset and test on the other (out-domain) dataset. CLIP (Radford et al., 2021) is pre-trained on 400 million image-text pairs using the contrastive learning method and has strong zero-shot generalization ability. By calculating feature similarity with a given image and candidate texts in a list, it realizes open-vocabulary image-text tasks. Therefore, on this basis, many methods in the field of text-to-image generation, a series of Diffusion-style-based and GPT-style-based methods (Rombach et al., 2022; Sun et al., 2024) are proposed to use CLIP to extract features to improve the generalization ability of the model. Based on this, MotionCLIP (Tevet et al., 2022a) proposes to fit the motion data of the training set to the mesh of the human SMPL (Loper et al., 2023) model in the preprocessing stage, thereby rendering multi-view static poses. In the training stage, the motion features are aligned with the text and image features extracted by CLIP simultaneously, thereby aligning the motion features to the CLIP space and enhancing the generalization ability of the model. AvatarCLIP (Hong et al., 2022) first synthesizes a key pose and then matches from the database and finally optimizes to the target motion. OOHGM (Lin et al., 2023), Make-An-Animation (Azadi et al., 2023), and PRO-Motion (Liu et al., 2023) first train a generative text-to-pose model with diverse text-pose pairs. Then, OOHGM further learns to reconstruct full motion from the masked motion. Inspired by the success of AnimateDiff (Guo et al., 2023), Make-An-Animation inserts and finetunes the temporal adaptor to achieve motion generation. PRO-Motion leverages the large language model to give the key-pose descriptions and synthesize motion by a trained interpolation network. However, all these methods, aligning the static poses with the image space, lose the temporal information during the learning and finally result in the generation of unrealistic motion. Recently, OMG (Liang et al., 2024a) try to use the successful paradigm, pretrained-then-finetuning, in the LLM to achieve open-vocabulary. Therefore, it first pretrained a un-condiditonal diffusion model with unannotated motion data, then finetunes on the annotated text-motion pairs by a ControlNet and MoE structure. However, due to the extremely limited labeled data, this type of method can ultimately only achieve the effective mapping from sub-text-space to sub-motion-space, which is still far from sufficient for achieving open-vocabulary tasks.

## 2.3 GENERATIVE MASK MODELING

BERT (Devlin, 2018) as a very representative work in the field of natural language processing, pretrains a text encoder by randomly masking words and predicting these masked words. Numerous subsequent methods in the generative fields have borrowed this idea to achieve text-to-image or video generation, such as MAGVIT (Yu et al., 2023), MAGE (Li et al., 2023), and Muse (Chang et al., 2023). Compared to autoregressive modeling, generative masked modeling has the advantage of faster inference speed. In the motion field, MoMask (Guo et al., 2024) first introduced generative masked modeling into the field of motion generation. It adopts a residual VQ-VAE (RVQ-VAE) and represents human motion as multi-layer discrete motion tokens with high-fidelity details. In the training stage, the motion tokens of the base layer and the residual layers are randomly masked by a masking transformer and predicted according to the text input. In the generation stage, the masking transformer starts from an empty sequence and iteratively fills in the missing tokens. In this paper, our overall architecture also adopts the similar generative masked modeling as MoMask (Guo et al., 2024) to implement our pretrain-then-finetuning strategy.

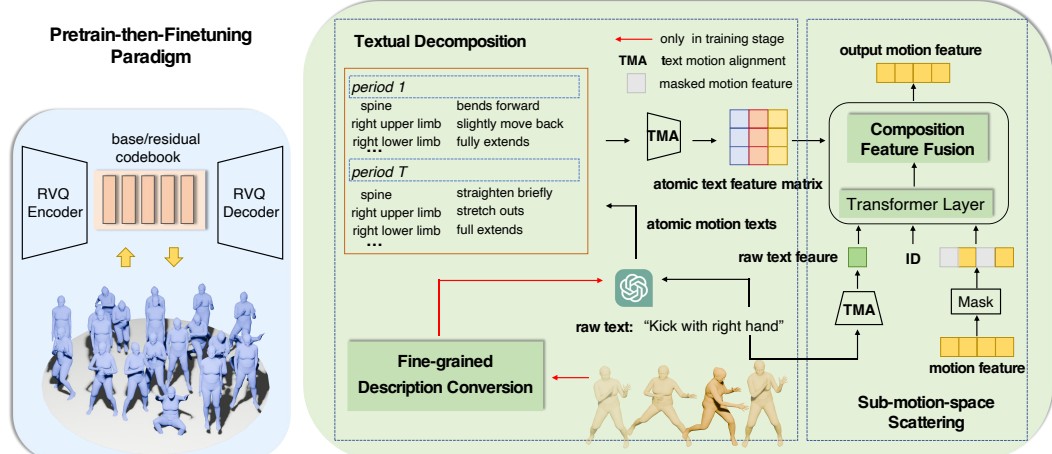

Figure 2: **The architecture of our entire framework**. The overall pipeline adopts discrete generative modeling. 1) In the Motion Pre-Training stage (left blue part), we use the Residual VQ-VAE (RVQ) model, which designs a base layer and $R$ residual layers to learn layer-wise codebooks. By tokenizing the motion sequence into multi-layer discrete tokens, we learn the large-scale motion priors. 2) In the Motion Fine-tuning stage (right green part), we first leverage the large language model(LLM) and the **fine-grained description conversion** algorithm we design (only used in training stage) to perform **textual decomposition**, which convert the raw text of a motion into the atomic texts. Then, for the base layer and residual layers in RVQ, we separately use generative mask modeling and a neural network with several Transformer layers to learn how to predict discrete motion tokens according to a given text. Furthermore, We design a text-motion alignment (TMA) module and a compositional feature fusion (CFF) module to learn the combinational process from atomic motions to the target motions.

## 3 METHOD

We propose a novel DSO-Net for open-vocabulary motion generation, aiming to generate a 3D human motion **x** from an open-vocabulary textual description **d** that is unseen in the training dataset. As shown in Fig. 2, our framework takes the pretrain-then-finetuning paradigm, which is first pretrained on a large-scale unannotated motion dataset, and then finetuned on a small dataset of text-motion pairs. All those motions are processed in a unified format by UniMocap (Chen & UniMocap, 2023). As analyzed before, the key to achieving open-vocabulary motion generation is to establish the alignment between the full-text-space and the full-motion-space, which we call full-mapping. To this end, we leverage atomic motions as intermediate representations and convert the full-mapping process into two orderly coupled stages: 1) Textual Decomposition, and 2) Sub-motion-space Scattering. The Textual Decomposition stage aims at converting any given motion text into several atomic motion texts, each describing the motion of a simple body part over a short time period, thereby mapping the full-text-space to the full-atomic-text-space. The Sub-motion-space Scattering stage is designed to learn the combinational process from atomic motions to the target motions, which scatters the sub-motion-space learned from limited paired data to form the full-motion-space, via a text-motion alignment (TMA) module and a compositional feature fusion (CFF) module. The scattered sub-motion-space eventually improves the generalization of our motion generation ability.

### 3.1 TEXTUAL DECOMPOSITION

The textual decomposition is designed to convert any given motion text into several atomic motion texts (each describing the motion of a simple body part in a short time period). Different from the raw motion texts that contain abstract and high-level semantic meanings, which hinder the model's ability to understand open-vocabulary motion texts, atomic motion texts provide a low-level and concrete description of different limb movements, which are shared across different domains. Therefore, we use the text of atomic motions as an intermediate representation, first converting raw

motion texts into atomic motion texts, and then learning the process of combining atomic motions to generate target motions.

First, we construct atomic motion and text pairs from a limited raw motion-text paired dataset. Although Large Language Models (LLM) have the generalization ability to describe any given motion as an atomic motion, directly inputting a raw motion text into a large model will lead to a mismatch between the generated description and the real motion behavior, where some works also said before He et al. (2023); Shi et al. (2023). For this reason, we design a **Fine-grained Description Conversion** algorithm to describe the movement of each joint and the relative movement relationship between joints in a fine-grained way for a given 3D motion. Then, this fine-grained description and the raw motion text are input into the LLM to summarize it into the final atomic motion description. The entire algorithm describes the movement of body parts in the input motion from three aspects: speed, amplitude, and specific behavior; and divide the entire movement into at most $P$ time periods. Specifically, this fine-grained description conversion consists of four steps:

**Pose Extraction.** For each frame in a given 3D motion $x_i$, we compute different **pose descriptors**, including the angle, orientation, and position of a single joint, and the distance between any two joints. Taking the angle as an example, we can use three joint coordinates, $J_{shoulder}$, $J_{elbow}$, and $J_{wrist}$, to compute the bending magnitude of the upper limb, which is formulated as:

$$\frac{J_{shoulder} - J_{elbow}}{||J_{shoulder} - J_{elbow}||} \odot \frac{J_{wrist} - J_{elbow}}{||J_{wrist} - J_{elbow}||},\tag{1}$$

where $\odot$ represents the inner product. For each frame in a motion, we compute pose descriptors for different body parts.

**Pose Aggregation.** After obtaining the pose descriptors of each frame in a motion, we aggregate adjacent frames into motion clips based on the pose descriptors, and obtain the descriptors of the motion clips. Given the pose descriptors $PD_{i-1}$, $PD_i$, and $PD_{i+1}$ of three consecutive frames, we first calculate the difference between two frames, $\Delta PD_{i-1} = PD_i - PD_{i-1}$ and $\Delta PD_i = PD_{i+1} - PD_i$. We determine whether these three consecutive frames should be merged into a motion clip based on whether the signs of $\Delta PD_i$ and $\Delta PD_{i+1}$ are the same, i.e., both positive or both negative. In this way, we start from time $i$ and continuously add the $\{\Delta PD_i, \Delta PD_{i+1}, \cdots\}$ with the same sign until the sign changes, and the result of addition is defined as $S_{PD_i} = \sum_{t=i}^{t=i+T_i} \Delta PD_t$ ($T_i$ is the consecutive time length from the starting time $i$), which represents the intensity change of the pose descriptor during the motion clip. We then calculate the velocity as $V_{PD_i} = \frac{|S_{PD_i}|}{T_i}$, where $|S_{PD_i}|$ is the absolute magnitude of $S_{PD_i}$. Finally, we obtain the **clip descriptor** for the motion clip starting from $PD_i$, which is defined as the (intensity change, velocity) pair: $CD_{PD_i} = (S_{PD_i}, V_{PD_i})$.

**Clip Aggregation.** Subsequently, we aggregate the motion clips based on the start time of its clip descriptor. We uniformly divide a motion into $P$ bins in time, and put each motion clip into a bin according to its start time. The number of the bins $P$ is set empirically. The motion clips that are put in the same bin will be regarded as co-occurring motion clips.

**Description Conversion.** We further classify the motion clips to some categories based on the intensity change and velocity in the clip descriptor, and convert it into the text description. For example, we first determine the behavior of a motion clip is "bending" or "extending" according to the clip descriptor $CD_{PDi}$ of angle, where the negative $S_{PD_i}$ means "bending" and vice versa. Then, when $|S_{PD_i}|$ exceeds a threshold, it will be classified as "significant"; while when the $V_{PD_i}$ is below some threshold, it will be classified as "slowly". Finally, the converted text is "Bending/Extending significantly slowly".

Through this fine-grained description conversion algorithm, we first divide the given motion into different several time periods (corresponding to $P$ bins), then the motion in each time period is converted to a fine-grained description composed of simple behaviors of body parts. Subsequently, we input all these fine-grained descriptions and the raw text of the corresponding motion into the LLM, and make it perform simplifications to summarize $L$ atomic texts for each period, where $L$ is the number of body parts, and each atomic text corresponds to a body part from spine, left/right upper/lower limbs, and trajectory. An example of the atomic motion texts is shown in Fig. 2-right. The atomic motion texts obtained by our algorithm are highly consistent with the real motion. During the inference, for any given motion text, we provide corresponding textual decomposition examples

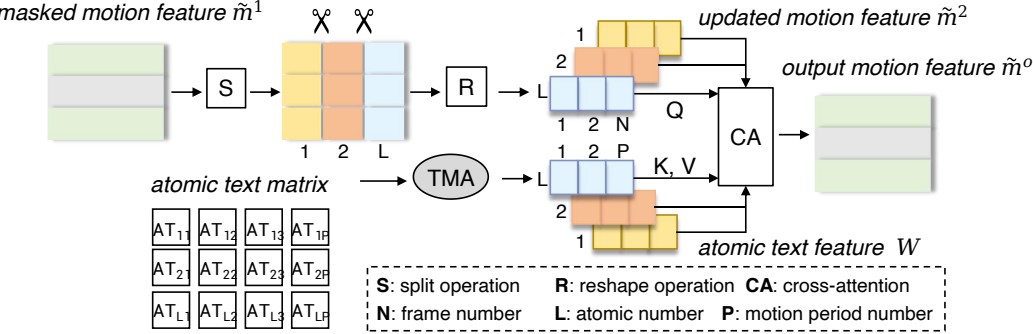

Figure 3: Details of the compositional feature fusion (CFF) module, where the atomic text matrix is input into the TMA module for feature extraction, and is fused with the motion feature by cross-attention.

to ask the LLM to decompose the motion into several periods, and decompose each period into $L$ atomic motion texts describing simple body part movements.

## 3.2 SUB-MOTION-SPACE SCATTERING

As mentioned before, given the limited paired data, we can only learn a **sub-motion-space**, i.e., a condensed subspace of the full-motion-space. To enhance the open-vocabulary generalization ability, we propose to learn the combination process from atomic motions to the target motion, thereby rearranging the sub-motion-space we learned in a much more scattered form. Although some target motions are out-of-distribution for the sub-motion-space, we **scatter the subspace to form the full-motion-space**, and convert the generative process from extrapolation into interpolation, thereby significantly enhancing the generative generalization ability of our model.

Specifically, to enable our network to learn a scattering sub-motion-space, we propose to establish the combination process of atomic motions instead of learning the target motion directly, which consists of two main parts: (1) Text-Motion Alignment (TMA) and (2) Compositional Feature Fusion (CFF).

**Text-Motion Alignment (TMA).** Previous T2M generation methods usually leverage the CLIP model to extract text features, and then align the text feature into the motion space through some linear layers or attention layers during training. However, the CLIP method is trained on large-scale text-image pair data, where the text is a description given for a static image, which has a huge gap with the description of motion (including dynamic information over a time period). As a result, learning the alignment during training brings an extra burden and seriously interferes with our network's focus on learning the combination process from atomic motions to the target motion. Therefore, inspired by the text-motion retrieval method TMR (Petrovich et al., 2023), we first use the contrastive learning method to pretrain a text feature extractor TMA on text-motion pair data. Compared to the CLIP encoder trained on text-image pair data, our TMA is trained on text-motion pair data, which better aligns the text features to the motion space. Specifically, we use the InfoNCE loss function to pull the positive pair $(x^+, d^+)$ (a motion and its corresponding textual description) closer, and push the negative pair $(x^+, d^-)$ (a motion and another textual description) away, to ensure that the text features are aligned with the motion space. For $M$ positive pairs, the loss function is defined as follows:

$$\mathcal{L}_{\text{NCE}} = -\frac{1}{2M} \sum_i \left( \log \frac{\exp A_{ii}/\tau}{\sum_j \exp A_{ij}/\tau} + \log \frac{\exp A_{ii}/\tau}{\sum_j \exp A_{ji}/\tau} \right), \quad (2)$$

where $A_{ij} = (m^+, d^+)$, and $\tau$ is the temperature hyperparameter. Subsequently, as shown in Fig. 2, we use TMA as our text feature extractor for both the raw texts and decomposed atomic texts. All these atomic text features are input to our next compositional feature fusion (CFF) module to guide the motion generation. In this way, we greatly reduce the interference caused by text-motion misalignment during the atomic motion combination learning process.

**Overall Architecture of Motion Generative Model.** Our motion generative network contains two different generative models, one corresponds to the base layer of Residual VQ-VAE (RVQ), and the other corresponds to the residual layers. As detailed in Fig. 2, each generative model, either base-layer or residual-layer, contains sequentially stacked transformer layer and Compositional Feature Fusion (CFF), i.e., 1 transformer layer followed by 1 CFF module, repeated for $K$ times. Different residual layers share the same parameters, but have different input indicators, i.e., $V$ residual layers correspond to indicator 1 to $V$. Since the indicators are the minor difference between the base and the residual layer generative model, we omit it for convenience.

Given a motion **x** of $F$ motion frames, we first sample it with ratio $r$, and encode each of $\frac{F}{r}$ down-sampled frames with the motion encoder in RVQ. We then perform quantization by mapping the encoded features to the code indices of their nearest codes in the codebook of base/residual layers, denoted as $I = [I_1, I_2, ...I_N]$, where $N = \frac{F}{r}$. Then, the code indices $I$ is first converted into a one-hot embedding, and then mapped into a motion embedding $m = [m_1, m_2, ...m_N] \in R^{N \times D_m}$ by linear layers, which is taken as the initial input to the generative models, where $D_m$ is the channel dimension.

Given a text, and the atomic description (a $L \times P$ text matrix), we encode the raw text and the atomic texts by our TMA text encoder, and the outputs are: 1) raw text feature $T_r \in R^{D_T}$, where $D_T$ is the channel dimension; and 2) atomic text feature $W \in R^{L \times P \times D_W}$, where $L$ is the number of atomic, $P$ is the number of motion periods, and $D_W$ is the channel dimension.

The motion embedding $m$ is first randomly masked out with a varying ratio, by replacing some tokens with a special [MASK] token. Subsequently, the masked embedding $\tilde{m} = [\tilde{m}_1, \tilde{m}_2, ...\tilde{m}_N]$ is combined with the raw text feature $T_r$, which is then input into a transformer layer $\mathcal{F}_{Transformer}$. The outputs are refined raw text feature and refined motion feature, denoted as $T_r^o$ and $\tilde{m}^1 = [\tilde{m}_1^1, \tilde{m}_2^1, ...\tilde{m}_N^1] \in R^{N \times D_m}$, which is formulated as:

$$T_r^o, \tilde{m}^1 = \mathcal{F}_{Transformer}(T_r; \tilde{m}). \tag{3}$$

The transformer layer enables the output $\tilde{m}^1$ to integrate both global information and the temporal relationship.

**Compositional Feature Fusion (CFF).** The CFF module is designed to fuse atomic text features into motion features, and guide the model to learn the combinational process from atomic motions to the target motions. As shown in Fig. 3, we utilize the cross-attention mechanism to fuse the atomic motion text feature $W$ into the motion feature in a spatial combination manner. Specifically, we split the refined motion feature $\tilde{m}^1$ into $L$ parts along the channel dimension ($L$ is the number of body parts), and reshape the splitted motion feature and input it into a linear layer to obtain the updated motion embedding $\tilde{m}^2 \in R^{L \times N \times D_W}$. Then, the $\tilde{m}^2$ is taken as the Query, and the the atomic text feature $W$ is taken as the Key and the Value to conduct the cross-attention calculation. Since the atomic text feature $W$ is extracted by our TMA model, which has aligned the text space with the motion space, the output motion feature of the cross-attention $\tilde{m}^3 \in R^{L \times N \times D_W}$ could composite the atomic motions explicitly and directly. The overall process of CFF module is formulated as:

$$\tilde{m}^3 = \mathcal{F}_{CFF}(\tilde{m}^2; W). \tag{4}$$

Eventually, the $\tilde{m}^3$ goes through a linear layer and is reshaped to the final output motion feature $\tilde{m}^o \in R^{N \times D_m}$. The $\tilde{m}^o$ is then combined with the refined raw text feature $T_r^o$ and input to the next transformer layer and the CFF module. The transformer layer and the CFF module are sequentially stacked for $K$ times. The output motion feature of the final CFF module is input into a classification head and punished by a cross-entropy loss.

### 3.3 Inference Process

During the inference stage, for generative models of the base layer and $R$ residual layers, we initialize all motion tokens as [MASK] tokens. During each inference step, we simultaneously predict all masked motion tokens, conditioned on both the raw motion text and the atomic motion texts using in-context learning. Our generative models first predict the probability distribution of tokens at the masked locations, and sample motion tokens according to the probability. Then the sampled tokens with the lower confidences are masked again for the next iteration. Finally, all the predicted tokens are decoded back to motion sequences by the RVQ decoder.

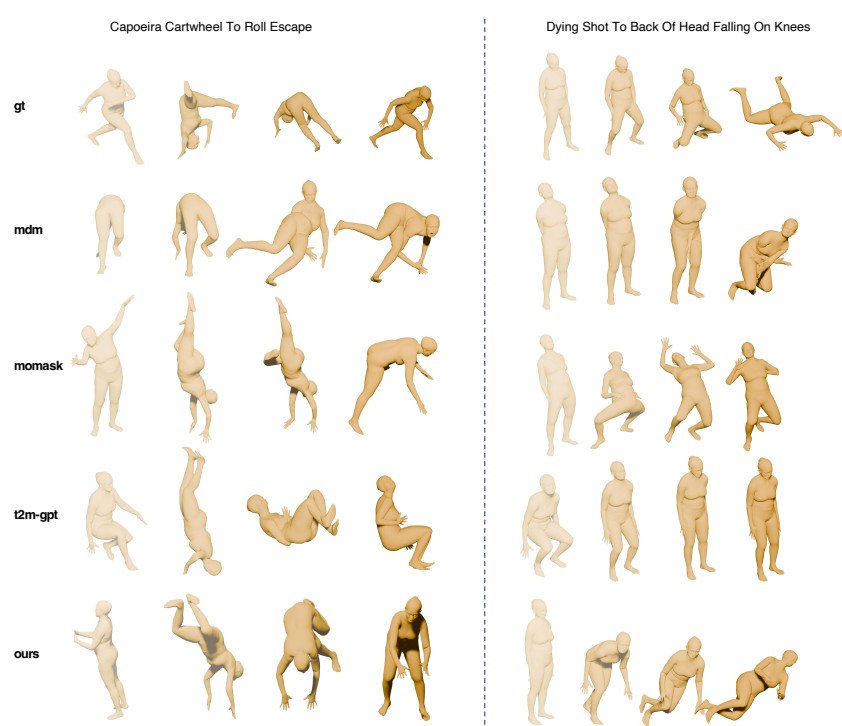

Figure 4: Comparison with several state-of-the-arts on open vocabulary texts.

# 4 EXPERIMENTS

## 4.1 EXPERIMENT SETUP

**Dataset Description.** In the pre-training stage, we utilize various publicly available human motion datasets, including MotionX (Lin et al., 2024), MoYo (Tripathi et al., 2023), InterHuman (Liang et al., 2024b), KIT-ML (Plappert et al., 2016), totaling over 22M frames. In the subsequent finetuning stage, we train our generative motion model using the text-motion HumanML3D dataset. During Inference, we experiment on three datasets, where one is the in-domain dataset (HumanML3D) while the other two are the out-domain datasets (Idea400 and Mixamo). Idea400 is the monocular dataset consisting of 400 daily motions, while the Mixamo includes various artist-created animations. The datasets used in pre-training only contain motion data and do not contain textual descriptions, while the datasets used in finetuning and inference contain motion data and annotated textual descriptions. Evaluation Metrics and Implementation Details are introduced in the appendix.

## 4.2 COMPARISON

We first compare our approach quantitatively with various state-of-the-art methods. From the Tab. 1, it can be seen that our method significantly outperforms the other methods on the two out-domain datasets (Idea400 and Mixamo) and also achieves comparable results on the in-domain datasets (HumanML3D). As shown in Fig. 4, compared with other representative methods, the qualitative generations of our model are much more consistent with the open-vocabulary texts. Both the quantitative and qualitative results fully illustrate that our two orderly coupled designs (textual decomposition and sub-motion-space scattering) enable the model not to overfit the distribution of a limited dataset but to possess strong generalization ability. Please check the appendix and the demo video for more visualization results on generated open vocabulary motion.

## 4.3 ABLATION STUDY

To examine the specific function of each module in our novel DSO-Net, we conduct a series of ablation studies focusing on the effect of pre-training on large-scale motion data, the effect of textual decomposition, the effect of text-motion alignment, and the effect of compositional feature fusion.

| Method | HumanML3D | | | Idea400 | | | Mixamo | | |
|---|---|---|---|---|---|---|---|---|---|
| | FID↓ | R-Prescion↑ | Diversity↑ | FID↓ | R-Prescion↑ | Diversity↑ | FID↓ | R-Prescion↑ | Diversity↑ |
| MDM | $0.061^{\pm.001}$ | $0.817^{\pm.005}$ | $1.380^{\pm.004}$ | $\mathbf{0.821}^{\pm.003}$ | $0.286^{\pm.006}$ | $1.329^{\pm.006}$ | $0.211^{\pm.002}$ | $0.380^{\pm.005}$ | $1.352^{\pm.005}$ |
| T2M-GPT | $0.013^{\pm.001}$ | $0.833^{\pm.002}$ | $1.382^{\pm.004}$ | $0.934^{\pm.002}$ | $0.314^{\pm.004}$ | $1.330^{\pm.004}$ | $0.221^{\pm.002}$ | $0.389^{\pm.004}$ | $1.350^{\pm.004}$ |
| MoMask | $\mathbf{0.011}^{\pm.001}$ | $0.878^{\pm.002}$ | $\mathbf{1.390}^{\pm.004}$ | $0.880^{\pm.001}$ | $0.340^{\pm.005}$ | $\mathbf{1.340}^{\pm.004}$ | $0.209^{\pm.001}$ | $0.415^{\pm.005}$ | $1.346^{\pm.005}$ |
| MotionCLIP | $0.082^{\pm.002}$ | $0.331^{\pm.002}$ | $1.281^{\pm.004}$ | $1.112^{\pm.002}$ | $0.237^{\pm.005}$ | $1.195^{\pm.007}$ | $0.3^{\pm.002}$ | $0.228^{\pm.004}$ | $1.176^{\pm.005}$ |
| Ours | $0.027^{\pm.002}$ | $\mathbf{0.957}^{\pm.002}$ | $\underline{1.388}^{\pm.004}$ | $\underline{0.847}^{\pm.001}$ | $\mathbf{0.703}^{\pm.004}$ | $\underline{1.338}^{\pm.004}$ | $\mathbf{0.186}^{\pm.001}$ | $\mathbf{0.807}^{\pm.004}$ | $\mathbf{1.360}^{\pm.004}$ |

Table 1: Comparison with state-of-the-arts on one in-domain dataset (HumanML3D) and two out-domain dataset (Idea400 and Mixamo).

| Methods | FID↓ | R-Precision↑ | | | Diversity↑ |
|---|---|---|---|---|---|
| | | R-Top1 | R-Top2 | R-Top3 | |
| Baseline | $0.898^{\pm.002}$ | $0.160^{\pm.004}$ | $0.251^{\pm.005}$ | $0.314^{\pm.004}$ | $1.342^{\pm.005}$ |
| Baseline+Pretrain | $0.890^{\pm.002}$ | $0.162^{\pm.003}$ | $0.256^{\pm.005}$ | $0.323^{\pm.004}$ | $1.340^{\pm.005}$ |
| Baseline+Pretrain + CFF | $0.886^{\pm.002}$ | $0.170^{\pm.004}$ | $0.266^{\pm.004}$ | $0.337^{\pm.006}$ | $1.333^{\pm.006}$ |
| Baseline+Pretrain + TMA | $\mathbf{0.844}^{\pm.002}$ | $0.380^{\pm.005}$ | $0.539^{\pm.005}$ | $0.630^{\pm.006}$ | $\mathbf{1.346}^{\pm.004}$ |
| Baseline+Pretrain + TMA + CFF | $\underline{0.847}^{\pm.001}$ | $\mathbf{0.449}^{\pm.006}$ | $\mathbf{0.613}^{\pm.004}$ | $\mathbf{0.703}^{\pm.004}$ | $1.338^{\pm.004}$ |

Table 2: Ablation Study on the Idea400 dataset. The TMA and CFF represent the text-motion-alignment module and the compositional feature fusion module.

**Effect of Pretraining on Large-scale Motion Data.** As we analyzed before, for open-vocabulary motion generation, we need to establish the mapping between text-full-space and motion-full-space. Therefore, to enlarge the motion space contained in our model, we leverage the large-scale unannotated motion data to pre-train a residual vq-vae. As illustrated in the second row in Tab. 2, we gain a 1% and 2% improvement in FID and R-Top3, which means enlarging the motion space learned in the model is useful.

**Effect of Compositional Feature Fusion (CFF).** By generally comparing the results of the third row and the second row, as well as the results of the last row and the fourth row, we can find that through the CFF module we designed, the consistency between the generated motion and the out-of-domain motion distributions(FID) and the similarity with their input text (R-Precision) on out-of-domain datasets are significantly improved. This fully proves that by splitting the motion feature channels and explicitly injecting the combination of atomic motions into the motion generation process, we can learn the combination process from atomic motions to target motions well and scatter the sub-motion-space we learned. Eventually, the scattered sub-motion-space can effectively convert the extrapolation generation process of out-of-domain motion into an interpolation generation process, thereby significantly improving the generalization of the model.

**Effect of Text Motion Alignment (TMA).** As shown in the Tab. 2, after leveraging the text encoder of TMA (row 4th), R-top3 almost doubles compared to the second row. Furthermore, we can see that using the CFF module on top of TMA, R-top3 increases by 11% (4th row v.s. 5th row), while the CFF module without TMA increases by only 3% (2nd row v.s. 3rd row). Both results demonstrate the text feature aligned with motion space indeed release the burden of learning the atomic motion composition process.

## 5 CONCLUSION

In this paper, we propose our **DSO**-Net, i.e. Textual **D**ecomposition then Sub-motion-space **S**cattering, to solve the **o**pen-vocabulary motion generation problem. Textual decomposition is first leveraged to convert the input raw text into atomic motion descriptions, which serve as our intermediate representations. Then, we learned the combination process from intermediate atomic motion to the target motion, which subsequently scattering the sub-motion-space we learned and transforming extrapolation into interpolation and significantly improve generalization. Numerous experiments are conducted to compare our approach with previous state-of-the-art approaches on various open-vocabulary datasets and achieve a significant improvement quantitatively and qualitatively.

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
