# OpenReview forum: "Textual $\textbf{D}$ecomposition then Sub-motion-space $\textbf{S}$cattering for $\textbf{O}$pen-Vocabulary Motion Generation"
_ICLR.cc/2025/Conference — ICLR 2025 Conference Withdrawn Submission_

### Official Review · Reviewer_2VBT · 2024-11-03

**Soundness:** 2
**Presentation:** 2
**Contribution:** 2
**Rating:** 3
**Confidence:** 5

**Summary:**

This paper proposes a text-to-motion framework, aimed at the open-vocabulary motion generation problem. Specifically, this paper proposes to learn a sub-motion space, using atomic motion representations as an intermediate representation, and leveraginhg textual decomposition and sub-motion space scattering for the action representation mapping learning. The proposed method achieves strong performance over existing baselines.

**Strengths:**

1. The motivation to decompose an action description into a set of atomic action is strong. Using LLMs to extract the atomic actions in the textual format has not been widely used before.
2. The proposed method achieves strong performances, especially the R-Precision that reflects motion-text alignment, on common benchmarks.

**Weaknesses:**

1. This paper is not well-written. One of the most fundamental problem is how the motion is generated and the training objectives are not described. Do the authors use diffusion models or VAE?
2. While the the concept of text-based atomic action decomposition is well-motivated, it lacks one of the most important comparison results, i.e., the comparison with CLIP-based alignment (top right of Fig. 1). Without this, it is unclear whether the text-based decomposition outperforms the decomposition in the latent space.
3. This paper lacks a direct comparison with the latent-space atomic action decomposition method [a], which also learns atomic actions using cross-attention. The authors should describe the differences and compare the performance.
4. The proposed method could not achieve strong FID on common benchmarks, indicating the generated motions are not realistic, which could be verified in some of the videos in the supplement.



[a] Zhai, Yuanhao, Mingzhen Huang, Tianyu Luan, Lu Dong, Ifeoma Nwogu, Siwei Lyu, David Doermann, and Junsong Yuan. "Language-guided human motion synthesis with atomic actions." In *Proceedings of the 31st ACM International Conference on Multimedia*, pp. 5262-5271. 2023.

**Questions:**

1. Can the authors verify why the text-based action decomposition outpeforms the decomposition in the latent-space?
2. Can the authors explain why the proposed method achieves strong R-Precision, but could not improve over FID?

---

### Official Review · Reviewer_unhk · 2024-11-04

**Soundness:** 2
**Presentation:** 2
**Contribution:** 2
**Rating:** 5
**Confidence:** 3

**Summary:**

In this paper, the authors proposed DSO-Net, an open-vocabulary text-to-motion generation framework. The proposed method achieve competitive performance across multiple benchmark.

**Strengths:**

1. The proposed method achieve competitive performance across multiple benchmark.

**Weaknesses:**

1. The proposed atomic motion idea has been discover in [1].
2. As claimed in the main paper that  human beings tend to partition it into the combination of several simple body part motions over a short time period, there are no quantitative or qualitative results to support the claim.
3. Missing metrics. MMDist and MModality are commonly used metric in text2motion task which are not included in this paper.


[1] Language-guided human motion synthesis with atomic actions, Zhai et al, ACM MM

**Questions:**

N/A

---

### Official Review · Reviewer_fQoT · 2024-11-07

**Soundness:** 3
**Presentation:** 3
**Contribution:** 2
**Rating:** 5
**Confidence:** 5

**Summary:**

This paper proposes a method for improving open-vocabulary motion generation. It consists of two main components, i.e.,  textual decomposition and sub-motion-space scattering, where the former uses LLM to generate the descriptions of atomic motions and the latter further learns the generative combination of atomic motions by a text-motion alignment (TMA) module and a compositional feature fusion (CFF) module.

**Strengths:**

1.  The idea of atomic actions has been explored by some existing works (see weakness). Specifically, the paper
decomposes the motion text into the texts of different body parts.

2. The paper is well-structured and provides a clear and detailed description of the proposed framework.

**Weaknesses:**

1 Limited novelty.  The idea of atomic texts/actions has been proposed in [A] and [B], but the authors did not cite these papers. The differences between the proposed method and them should be fully discussed.

[A] Language-guided Human Motion Synthesis with Atomic Actions, ACM MM 2023

[B] Generative Action Description Prompts for Skeleton-based Action Recognition, ICCV 2023

2 Insufficient experiments.  The paper fails to compare the proposed motion generation framework with some important SOTA  methods, e.g. [C], [D]…

[C] AttT2M: Text-Driven Human Motion Generation with Multi-Perspective Attention Mechanism, ICCV 2023

[D] MMM: Generative Masked Motion Model, CVPR 2024

**Questions:**

1 It would be helpful to compare with more state-of-the-art motion generation algorithms.

2 Could you please provide a brief discussion on the computational complexity, which is important to real-time applications? And discuss any limitations or potential pitfalls in the proposed methods?

---

### Note · Authors · 2024-11-13

**Comment:**

withdrawal improves this work.

**Withdrawal Confirmation:**

I have read and agree with the venue's withdrawal policy on behalf of myself and my co-authors.